# The Gospel's Double Antisymmetry as the End-Point of the Development of Western Society

**Borut Pohar**

Faculty of Theology, University of Ljubljana, 1000 Ljubljana, Slovenia; borut.pohar@teof.uni-lj.si

**Abstract:** In this article, we addressed the question of whether the Gospel's double antisymmetry can be found in reality as such and especially in Western society, which has obviously made developments in its history. Based on the conceptual analysis of language, we came to the conclusion that there are four layers of reality: the material world, lifeworld, material life and personal life. Through the analysis of individual layers of reality, we have come to the realization that they are characterized by the double antisymmetry (horizontal and vertical) spoken of in the Gospel. While the world (material and lived) is characterized by the primacy of parts over the whole, life (material and personal) is characterized by the primacy of the whole over the parts. Furthermore, if the material world and material life are characterized by the supremacy of the abstract over the concrete, the lifeworld and personal life are characterized by the supremacy of the concrete over the abstract. Based on examples from logic, science, and society, we also show how this antisymmetry manifests itself in concrete phenomena of everyday life and how it points to the gradual conversion of Western society, which is, according to our conclusion, becoming more and more like the People of God, the heir of the Kingdom of Heaven proclaimed by Jesus of Nazareth.

**Keywords:** Gospel; double antisymmetry; mountain-body model of understanding; four-layered reality; material world; lifeworld; material life; personal life; science; Western society; conversion

## 1. Introduction

The article draws inspiration from Alister E. McGrath, as it follows from the principles of his critical realism and his vision of Trinitarian natural theology, which is based on the explanatory capaciousness of the Christian faith. It is an attempt to define a big picture of Reality, the Christian ''map of meaning'' or framework, which is hidden in the depths of Christian tradition, and to show that it makes sense of our observations and experience (McGrath 2009, chp. 5). In fact, this is a project that goes beyond the scope of this article, in which we focus on only one part of Reality, namely the phenomenon of Western society and try to find its big picture, i.e., the content of all of its four ontological layers. In doing so, we introduced many new ideas and concepts that, unfortunately due to lack of space, we could not develop in full detail. In line with the principles of critical rationalism, we have decided to put a part of this emerging project into critical public debate in order to eliminate possible mistakes and make up for shortcomings from the very beginning.

In line with the outlined vision of Trinitarian natural theology, our approach is holistic, which means that we reject the approach to philosophy that comes from the German Research University model. Namely, we are not searching for some small 'holes' in the published record or in the totality of knowledge. Rather, we are attempting a synthetic and synoptic approach, which connects the fragmented and disordered parts of our knowledge into a meaningful whole. This theoretical framework gives a new way of looking at reality as a whole, and in particular a new way of understanding that emerges from our interdisciplinary approach. We are not just looking for a map of reality as a whole, but also for the most well-trodden theoretical path that connects the point of departure and the goal, which are depicted on the map.

The purpose of the article is to give an optimistic perspective on the direction of development of Western society, which is often pessimistically said to be in a decadent state or threatened with collapse, something which was announced more than a hundred years ago by Oswald Spengler in his book *The Decline of the West*. Today this view is defended by authors such as the Iranian-Canadian philosopher Ramin Jahanbegloo, who in his 2017 work, *The Decline of Civilization*, argues that the West and the world at large are in the process of 'de-civilisation' because people are more and more senseless and without empathy. American political scientist Fukuyama (1992, p. 289) in his book *The End of History and the Last Man* similarly argues that humanity with the victory of the liberal–democratic model of society over the Soviet system has reached the end point of history as such, for there can be no further progress to an alternative system.

On the basis of a theoretical analogous explanation of the development of Western society and its principles, we claim that the development of the West has not stopped historically, nor is it declining, but instead it is advancing towards abandoning worldliness and enforcing the eductive logic of personal life that is slowly coming to the surface in Western society. We argue that this eductive logic is in fact the Gospel's logic of the Kingdom of God proclaimed by Jesus, who compared it to leaven (Luke 13:21), which means it is slowly changing society for the better. Jesus also commanded his disciples to be the salt of the earth (Matthew 5:13), and so to give the world in which they live a taste for love, for the main commandment of the Kingdom of God is a commandment of love, which includes love of enemies (Matthew 5:44) and the poor (Matthew 5:7). Our hypothesis thus says that throughout history, Western society has been steadily converting in the direction of evangelical double antisymmetry. The Greek word "*metanoia*" also means "a change of mind," and in theology, a radical conversion that stems from the biblical God's requirement to abandon idolatry and to have faith in one God. "This demand dictates a radical conversion, a change of being, a new form of life. The God who reveals himself in the Pentateuch is uncompromising. He chose his people; mankind chooses him. Covenant is mutual election" (Petkovšek 2017, p. 624).

*Metanoia* is therefore not tied only to the individual, but to the whole people, who completely reject paganism and choose one God as their only God. *Metanoia* happens when a people becomes the People of God. According to our theory, two changes in the way of thinking must be made for any society to move from worldliness to eternal life, and we argue that these two changes sum up the evangelical spirit: the precedence of the whole over the parts and the supremacy of the concrete over the abstract. The precedence of the whole over parts means that people share a common way of life and mission and are thus organically connected and feel with each other. The supremacy of the concrete over the abstract, however, means that the Law of the People of God must not be to the detriment of any member of the communion, which means that no one is overlooked and that every voice is heard and taken into account.

The main bulk of the paper is about what this Gospel's double antisymmetry concretely means, how it manifests itself in everyday phenomena such as the structure of an argument, science and society, especially how the development of logic, science and society leads to the Gospel's logic, which is characteristic of personal life, on which the Kingdom of God is based.

Christianity as a religion therefore carries with it the requirement of conversion in order for the people to become the People of God, and conversion requires a double reversal in the way of thinking, which is presented in our article under the term double evangelical antisymmetry of personal life with respect to the worldly way of life. Thus, this article is about religion.

In this paper, we present our findings, that Western society has moved from an initial deductive pattern of thinking through an inductive and abductive phase to an eductive pattern of thinking. This means that Western society throughout its historical development has been characterized by a conversion in the direction of the Gospel's double antisymmetry,

in which the whole takes precedence over the part, and the abstract is subordinate to the concrete.

Our basic principle is the idea of four-layered reality. We argue that natural phenomena are stratified which means that we can identify four ontological layers in each. In this way, we obtain a full picture of the phenomena. In our article on different meaning levels of evolution (Pohar 2022, forthcoming) we analyze evolution as a state, event, happening, and identification. We will use this example of evolution to show how each phenomenon has four ontological layers. These layers are as follows:

(1)   A natural phenomenon as an immovable principled state of affairs is part of the external material world, which is outside and can be observed with the five senses. From these observations, principles, universal rules, scientific laws and causes can be deduced. If we take evolution as an example, we can say that living beings are in principle in a state of adaptedness to their environment, and their organs are in a state of vestigiality.

(2)   A natural phenomenon as a purposeful event is part of the inner lived world or lifeworld, which is the flesh of the world. It is the world's interior that we cannot observe but nevertheless can experience. From the experience thus obtained, we induce general laws (in the case of evolution, the law of adaptation), laws of nature (in the case of evolution, adaptability), and essences (in the case of evolution, adaptedness).

(3)   A natural phenomenon as an impersonal meaningful happening is part of the material life, which is the heart of nature. Phenomenologist Michel Henry (cf. Henry 2003) wrote about life as an independent reality that can be opposed to the world because he found that life and the world are significantly different from each other. We will define material life as an impersonal meaningful happening in which the whole takes precedence over the parts—as opposed to a world where parts take precedence over the whole—whereby this happening has an end, as it leads to a solution to a theoretical problem. This happening is explained by theoretical explanations, and the scientists who discover them are the problem solvers. (cf. Simon 1989, chp. 14). The theory of photosynthesis explains the happening of the production of glucose, which is the basic metabolite. With this mechanism, a plant solves its life problem of nutrient deficiency. This life is material because it is based on the dynamics of material entities such as proteins, which cooperate in perfect harmony. Similarly, the evolutionary mechanism of the struggle for existence solves the problem of maladaptation of organisms. It involves impersonal material happening as, for example, the struggle of animals with each other over limited food resources.

(4)   A natural phenomenon as a personal original identification is part of personal life, which is the innermost part of nature. Original identification means that an individual personal origin, e.g., mother, father, doctor, evolutionary biologist, etc., performs his profession in a dynamic way, for example by conducting research of the phenomenon of evolution. Thus, he or she sacramentally re-presents the phenomenon in a unique way, and at the same time realizes his or her identity. The individual profession and the exercise of the profession is thus the deepest part of reality, the true identity being written in the natural law, which is written in the human heart. This original identification is not random but has an end in the solution of personal (ethical) problems of mankind. If we lived according to natural law, there would be no personal problems. Personal life thus separates itself from the material not only by the nature of its dynamics (happening vs. identification), but also by the kind of problem it solves (theoretical vs. ethical)—as the title of the book by the philosopher of science Karl Popper says (1991), *All life is problem solving*. While the world creates problems with its worldliness, life solves them.

## 2. Methods

The ideas presented in this article, including the thesis that reality is four-layered, were reached on the basis of the following methods:

### 2.1. The Methods of Discovery

Based on the deductive abstraction (universalization) of Darwin's method of discovering the theory of evolution, which we deal in our aforementioned article (cf. Pohar 2022), we have identified the following methodological steps in the analysis of natural phenomena, including the phenomenon of Western society.

(A) First, we were aware of our research starting points. Observations of the material world are the starting point of our understanding of nature. On the basis of observations and their deductive abstraction, we discover the principles that govern nature and the causes that lead to the state of affairs which we observe. The natural phenomenon we observed was Western society.

Our starting point for understanding the phenomenon of Western society was the principles of Catholic doctrine (especially eschatology, which is the doctrine of the last things, including the messianic era and the Kingdom of God, through which history is to end), and the philosophical principles of personalism, which is "an approach or system of thought which regards or tends to regard the person as the ultimate explanatory, epistemological, ontological, and axiological principle of all reality" (Zalta 2020, s.v. "Personalism").

In addition, we also proceeded from the already established principles of critical realism, which is "one form of realism which has received especial attention in the dialogue between science and religion" (McGrath 2010, p. 78). These principles are as follows:

- The principle of the realism about ontology, which says that "a real world exists, external to the human mind, which the human mind can encounter, understand, and represent, if only in part" (McGrath 2010, p. 77).
- The principle of active involvement of the subject in the process of knowledge formation, which includes the principle of incomplete and insufficient mirroring of reality in language, the principle of fallibility of all knowledge and the principle of theory-ladenness of observation. Language, on the one hand, mirrors reality (Klaasse 2018, p. 133; Wittgenstein 1922, §4.01), which is "pregnant" with its form, i.e., with meaning (Merleau-Ponty 1964, p. 12); on the other hand, it also shapes our knowledge about it because "the human mind is active in the process of perception" (McGrath 2010, p. 78). For critical realists, "language does refer and represent—albeit inadequately—parts of reality" (Klaasse 2018, p. 14) with the help of mental models, such as metaphors and analogies, which are linguistic constructs. These constructs "do not exhaustively represent reality because there is always the possibility of being mistaken." (p. 14). Knowledge is thus always fallible. One of the reasons for its fallibility is the principle of theory-ladeness of observation because, according to critical realists, "there is no theory-free or theory-neutral knowing; all knowledge is theory-laden, due to the conceptual framework involved." (p. 14). We were aware that the content of our article might be influenced by the Christian worldview to which the author belongs.
- The principle of stratification of reality. According to Alister McGrath, there exists "a plurality of levels within reality, each demanding its own distinct mode of investigation and representation. Ontology determines epistemology." (McGrath 2003, p. 82). The principle of stratification is the basis of the universal rule that if we encounter any natural phenomenon, it must be composed of several ontological layers.
- The principle "ontology determines epistemology" has led us to the conclusion that for a particular logic or pattern of thinking (deductive, inductive, abductive, eductive), which is expressed, among other things, in different schemes of reasoning (e.g., *Modus ponens*), and which is connected to certain epistemic issues (e.g., the problem of induction), there must be a certain corresponding ontology. This means that different patterns of thinking are connected to different ontological layers of the Western society phenomenon, which also determine their nature. Things from a particular layer of Western society (deductive, inductive, abductive or eductive layer) are thus supposed to bear traces of an individual pattern of thinking. This prompted us to look for certain concrete examples of individual layers of Western society (as stated below, the principle of four-layered reality was discovered by conceptual analysis) from logic,

science and society, for every natural phenomenon manifests itself through different things, which belong to a certain layer of reality. By deductive abstraction we came to the universal rule, which says that if we encounter a thing, which belongs to human society (e.g., a form of social order), it must be the expression of one of four types of logic: deductive, inductive, abductive, or eductive. In fact, on the basis of the principle 'ontology determines epistemology', we inferred the existence of eductive logic on the basis of the existence of the fourth layer of reality, i.e., personal life, which did not correspond to any of the traditional patterns of thinking (deductive, inductive and abductive). This kind of filling the hole in scientific knowledge is a regular scientific practice, for it led, for example, to the discovery of new elements of the periodic table, the existence of which hinted at empty space in the table.

- The principles that are originally ours and which are the fruit of deductive abstraction are as follows:
- The principle of four-layered reality, which we discovered on the basis of the conceptual analysis of Slovenian language. We used this method already in our Mountain-body model of understanding (cf. Pohar 2021), which is the basis of this article. Living language contains a very rich epistemological terminology and a sedimented life wisdom. We could say that "phenomenologists and conceptual analysts are endeavouring to expose underlying structures (be they existential or linguistic), which are in some sense 'always already' implicitly known" (D'oro and Overgaard 2017, p. 8). In our case, we paid full attention to these subtle semantic differences between the concepts, and the analysis revealed a meaningful pattern, namely the four-layered reality. The following is a table (Table 1) with the examples of concepts which we classified under a certain layer of reality, which means that language itself indicates the existence of four layers of reality.

**Table 1.** Four different layers of reality and its key concepts.

| No. | Layer of Reality | 1. Material World | 2. Lifeworld |
|---|---|---|---|
| 1 | form of affiliation | with a body | with the soul |
| 2 | nature of a layer | state of affairs | event |
| 3 | type of antisymmetry | starting point | vertical antisymmetry |
| 4 | the whole | one body composed of a multitude of constituents (components) | uniform framework (structure, building) built from a multiplicity of building block |
| 5 | method of discovery | deductive abstraction (universalization) | inductive abstraction (generalization) |
| 6 | type of discovered meaning | principles, universal rules, scientific laws and causes | essences, laws of nature and general laws |
| 7 | basic argument for justification of a hypothesis | Modus ponens | inductive generalization |
| 8 | the strength of the argument | certainty | probability |
| 9 | type of civil society | autocracy | democracy |
| 10 | abstract/concrete | universal/particular | general/special |
| No. | Layer of Reality | 3. Material Life | 4. Personal Life |
| 1 | form of affiliation | with the heart | with the spirit |
| 2 | nature of a layer | happening | identification |
| 3 | type of antisymmetry | horizontal antisymmetry | double-antisymmetry |

**Table 1.** *Cont.*

| No. | Layer of Reality | 3. Material Life | 4. Personal Life |
|---|---|---|---|
| 4 | the whole | plurality of segments united in a single organism | multitude of members unified in a single spiritual body |
| 5 | method of discovery | abductive contextualization | eductive identification |
| 6 | type of discovered meaning | meaningful explanations and meaningful life's guides | original and purposive truths (identities) |
| 7 | basic argument for justification of a hypothesis | Affirming the consequent | eductive identification |
| 8 | the strength of the argument | possibility | maybe |
| 9 | type of civil society | constitutional state | Kingdom of God |
| 10 | abstract/concrete | common/single | communal/individual |

- We also discovered the principle of the Gospel's double antisymmetry with respect to the material world, to which the principle of worldliness applies. We discovered both principles and the difference between them on the basis of identifying essential differences between individual layers of reality, based on the analysis of certain cases listed in this article. First, we found out that we can arrange different things under logic (types of arguments), science (history of science, methods), and society (different forms of social order) among the four layers of reality, discovered by our conceptual analysis. Then we wondered how these cases from individual layers differ significantly from each other, using the inductive phenomenological method of eidetic reduction, which is a method for finding essential structures (cf. Zahavi 2019, p. 45). The analysis first showed that the layers differ significantly from each other in giving preference to either the whole or the part and in the type of subordination, namely whether the concrete is subordinate to the abstract or vice versa. We analyzed the relationship between a part and the whole and between the abstract and the concrete in an individual layer of Reality in even more detail, and it turned out that language takes these subtle differences into account, as we have specific expressions for each layer of Reality for the part, the whole, the abstract and the concrete in Slovene as well as in English (Table 1, no. 4). However, the analysis also showed that there is a double Gospel's antisymmetry between the given cases from personal life and the material world, and we changed this finding on the basis of deductive abstraction into a universal rule that in principle this is always the case, although we were fully aware that examining new cases may prove that such universalization was too hasty.

(B) Second, with the help of the inductive phenomenological method of eidetic reduction we identified conversion as the essence of the Western society (which is also its goal to be reached). We discovered that each layer of Western society (deductive, inductive, abductive and eductive) is in essence a certain degree of its conversion. There is a clear gradual transition from the deductive logic of the material world to the inductive logic of lifeworld, the transition from the inductive logic to the abductive logic of the material life, and the transition from the abductive logic to the eductive logic of personal life, which is the end-point of development. The event of transition of Western society from one logic or pattern of thinking to another was interpreted as conversion based on the discovered principle of evangelical double antisymmetry because with each transition, the society approached the logic of personal life for one step. The discovered essence of the conversion of Western society is in resonance with Christian eschatology, which prophetically predicts that people will one day become the People of God, which means that they will be converted as a whole, not just as individuals.

(C) Third, based on the method of abductive contextualization, which "is the only logical operation which introduces any new idea" (Peirce 1931), we discovered a theoretical explanatory pattern of understanding of Western society in three steps. In the first step, the analysis of the dynamics of the four layers of Western society revealed that there is a historical movement from the surface (worldly way of thinking) to the core (Gospel's way of thinking) and from the core to the surface because the principle of the personal life's double antisymmetry is slowly gaining a foothold in the material world of Western society, thus changing it; meanwhile, the material world's principle of worldliness is slowly sinking into oblivion. The second step was the introduction of Karl Rahner's principle of the symbolism of any Reality, which says that all reality is symbolic, which means that one reality (e.g., planet Earth) can refer to the other (e.g., nature as such or Western society) and call attention to it and express itself in it (cf. Rahner 1974, pp. 222–25). The third step in discovery of the theoretical explanation was the use of abductive contextualization method, which generated the abductive Earth's global dynamics analogical model of the development of the Western society. This model takes a dynamic interaction between the four layers of the planet Earth as an analog model mechanism of the interaction between our proposed four layers of Western society because both dynamics coincide in a meaningful way. Earth's inner core, outer core, mantle and crust are in a constant motion, where the interior comes to the surface and the surface partially sinks and is thus constantly changing. The same thing happens in the case of the four layers of Western society. In this way, we also gave meaning to our chosen phenomena of logic, science and society, as we gave them an explanation or a theoretical framework which explains the difference between them and connects them in a meaningful whole. Of course, this is just an abductive hypothesis that still needs to be tested.

### 2.2. The Methods of Justification

There are four basic ways in which we justify our discoveries.

(A) The basic principle of justification is language itself, with its sedimented meaning or wisdom. When we used the conceptual analysis of the Slovenian and English languages, we presupposed that language mirrors four layers of reality—albeit inadequately. As we mentioned before, critical realism is a fallibilist position.

(B) Second, we corroborated our thesis of four-layered reality with incorporating it in the abductive Earth's global dynamics analogical model of the development of Western society, which was tested with the inductive Method of the Inference to the Best Explanation. According to Alan Musgrave (2010, p. 93), this method can be represented by the following scheme:

Premise 1: It is reasonable to believe that the best available explanation of any fact is true.

Premise 2: F is a fact.

Premise 3: Hypothesis H explains F.

Premise 4: No available competing hypothesis explains F as well as H does.

Conclusion: Therefore, it is reasonable to believe that H is true.

With our analogical model, we made sense of certain examples from logic, science and society, and we claim that our theory is their best explanation, whereby "being a potential explanation of some evidence confirms a hypothesis" (Bird 2011, p. 276), and the best explanation is the one that "makes the most sense of what we actually observe in nature" (McGrath 2012, p. 85). In this case, we used the theoretical value empirical fit.

One of the theoretical values which is also used for the evaluation of theories is the power of unification. With explanatory unification, we try "to find a framework big enough to accommodate as much as possible" (McGrath 2012, p. 86). Our theory has proven to make sense of the separate areas of human knowledge, such as science, theology, or philosophy, which is an indicator of its veracity.

(C) Third, we took the principle of antisymmetry not only as a means to interpret the development of Western society as a conversion in the direction of evangelical double

antisymmetry, but also as evidence for the veracity of the principle of the four-layered reality because in science this is often the case.

The concept of symmetry can be found in practically all areas of human activity: architecture, biology, mathematics, music, physics, and other related topics. Symmetry, beauty, and truth are inextricably linked. First, people have always connected symmetry with beauty. Herman Weyl, in this sense, says this: "Symmetry, as wide or narrow as you may define its meaning, is one idea by which man through the ages has tried to comprehend and create order, beauty, and perfection" (Weyl 1952, p. 5). Furthermore, in science, there are people who see the connection between beauty and truth. Alister McGrath asserts that there is "deep-seated intuition within the scientific community that beauty is indeed a guide to truth, even if the mechanism and validation of this relationship is unclear" (McGrath 2008, p. 276). However, in science, symmetry is not only important for the context of justification but also for the context of discovery because "the pursuit of uncovering symmetry-related structures is a dominant research strategy in modern physics" (Zaidel and Hessamian 2010, p. 137), which is especially true in the case of laws of nature. Gross states (Gross 1996) this: "When searching for new and more fundamental laws of nature we should search for new symmetries" (p. 14259).

Symmetry is defined as "an invariance of an object or system to a set of changes (transformations)" (Hill and Lederman 2000, p. 349). In geometry, for example, this means that "when the two halves of a bilaterally symmetric figure are exchanged by reflection, we recover the original figure, and that figure is said to be invariant under left-right reflections" (Zalta 2021, s.v. "Symmetry and Symmetry Breaking."). In the case of the laws of nature, symmetry means their universal applicability regardless of place or time in space, which means that their formulas are the same everywhere and always, meaning that they are invariant under translations in space and time (Hill and Lederman 2000, pp. 349–51).

In nature, besides ordinary symmetry, we can also find antisymmetry, which is a specific type of symmetry also called "two-color" symmetry (Figure 1). Atoms, for example, possess charge and spin, which can be thought of as a kind of "color". For two states of charge, + and -, and for two states of quantum spin, ↑ and ↓, two different colors are used to represent them (e.g., black and white). "An antisymmetry operation switches a black characteristic into a white characteristic and vice versa" (Padmanabhan et al. 2020, pp. 256–58).

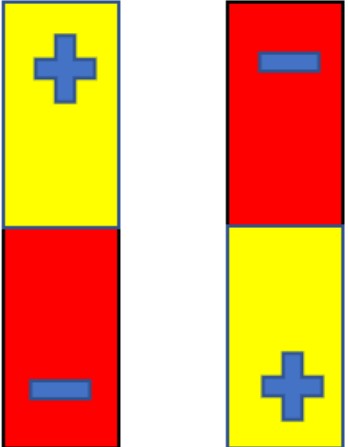

**Figure 1.** Antisymmetric combination.

We want to use this intuitive idea of the connection between symmetry and truth in the case of philosophy, which, unlike physics, is a universal science and deals with everything that exists. "The material object of philosophy is therefore the whole reality" (Hlebš 2016, p. 13).

In this article, we demonstrate that reality as a whole (material word, lifeworld, material life and personal life) is characterized by a horizontal and vertical antisymmetry, which we have named "Gospel's double antisymmetry" (Figure 2).

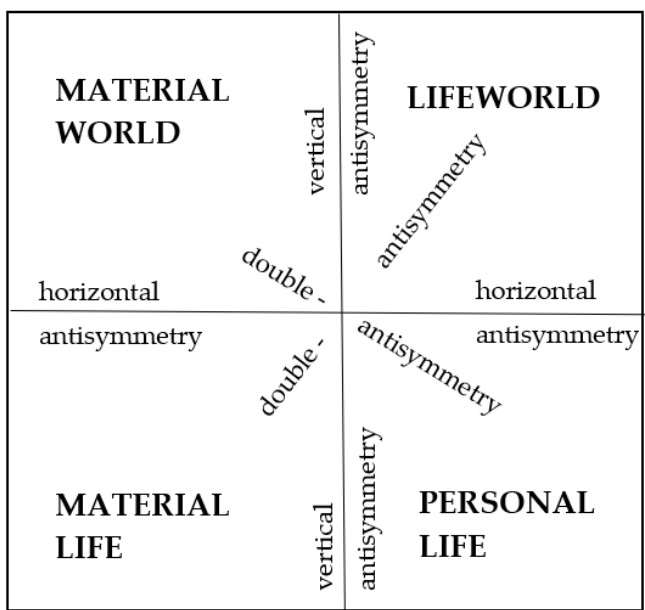

**Figure 2.** Four layers of Reality and their antisymmetric relations.

This double antisymmetry involves a translation from the first to the last place and from the highest to the lowest position, by which the importance of the object remains invariant. The starting point of our reflection or the origin of the idea was the Gospel's good news that for God, the poor and powerless people are of the same importance as the rich and powerful are for the world. The Gospel's hierarchy is clearly antisymmetrical to the traditional worldly Great Chain of Being.

In the Gospel, Jesus emphasizes two different dimensions of the translation: horizontal, in which the first and the last places are replaced, and vertical, in which the highest and the lowest position are replaced. In his words: "For behold, some are last who will be first, and some are first who will be last." (Luke 13:30); "For everyone who exalts himself will be humbled, but the one who humbles himself will be exalted." (Luke 14:11). This logic pervades the entire Gospel, therefore, we cite these words of Jesus as a summary of Gospel's pattern of thinking.

We can distinguish between two different polar domains of reality, the world and life, between which we can observe horizontal antisymmetry. Within the world and within life, there is also a vertical antisymmetry. In this article, we claim that each type of antisymmetry gives each domain of reality its characteristic nature, which is expressed in metaphysical characteristics, logical peculiarities, in science, and social order.

### 3. The Demonstration of the Gradual Conversion of the Western Society from the Logic of the Material World towards the Logic of the Personal Life

Western society is therefore a natural phenomenon that is stratified into four different layers, which belong to four different layers of Reality as a whole. These layers of Reality are: the external layer of the material world, the internal layer of the lived world, the outer core of the material life and the inner core of the personal life. We can therefore expect that the ontological peculiarities of each layer of Reality will be reflected in the phenomenon of Western society, which is part of this Reality, especially in its logic or pattern of thinking. We discovered that each layer of Western society is characterized by one of four logics of reasoning: deductive, inductive, abductive, and eductive, and that in each historical period of Western society one of these four logics came to the fore, which means that throughout

history Western society has manifested itself through ontologically different things. Rather, in each cultural period, things of a certain ontological type predominated (for example, in the first period of human history, a deductive autocratic form of government prevailed).

Each of these four logics has its own basic schema of reasoning, which in logic also serves as an argument to justify a particular statement (thesis). The reasoning schema is the form of the argument (Uršič and Markič 2009, p. 160). Below we will use these four types of arguments to justify different theses.

In what follows, we will present four different stages in the development of Western society, and at the same time we will show how Western society is gradually evolving toward eductive logic or the Gospel's double antisymmetry. We will do this by analyzing what in a certain layer of Reality occupies the first and last place and the highest and lowest position and what kind of antisymmetry this arrangement creates.

### 3.1. The Initial, Deductive Phase of Western Society

The deductive pattern of thinking that marked Antiquity and the Middle Ages originates in the material world. The material world of corporeality is the starting point of our understanding of reality, for without observation there can be no other deeper perceptions of meaning. We belong to the material world with our bodies, and we perceive the basic principled meaning in it by observation and further grasp it with the help of the rational deductive intuitive abstraction (universalization).

The material world is reflected in the deductive pattern of thinking, which is the main characteristics of the first stage of the Western society. The convenient symbol of the deductive logic is ancient Greece because it is the birthplace of the philosopher Aristotle, the father of deductive logic, as well as of the term autocracy, Kratos being the Greek personification of authority. The ancient Greek society is considered as the dawn of Western society. The deductive pattern of thinking prevailed in the West until the end of the Middle Ages. This means, in concrete terms, that people in all aspects of thinking gave priority to parts over the whole, and that they considered the concrete to be subordinate to the abstract.

3.1.1. The First and the Last Place: The Precedence of Parts (as Components) over the Whole (as a Body)

In the material world, a part is more important than a whole, which means that a part occupies the first place and a whole the last place. The material world is thus characterized by oneness in the multitude, that is, one body composed of a multitude of constituents (components), e.g., atoms. The whole is nothing more than the sum of the individual parts, which means that the components can exist independently (i.e., separate from the whole they compose). For example, if we tear a sheet of paper from a notebook, the sheet as a separate part still exists on its own, and so does the notebook as a whole. In this case, the whole and its parts are only loosely connected with each other, which means we cannot infer the properties of the whole from the properties of an individual component; for example, it is not possible to reason from the fact that the trees are small to the conclusion that the forest is also small.

(A) In logic, the deductive precedence of parts over the whole manifests itself in the rational deductive method of discovery, namely in the intuitive abstraction, used in philosophizing by famous ancient Greek philosophers such as Aristotle, who named it *epagoge*, as well as by medieval thinkers such as Thomas Aquinas. Intuition can be conceived as a non-analytical, nonconscious, unintentional, automatic, fast, effortless and experience-based thought process that results in a hypothesis or hunch (Zander et al. 2016, pp. 2–3). Experientiality of intuition means that one is improving in producing correct hypotheses or hunches that initiate action on the basis of acquired tacit knowledge (Zander et al. 2016, p. 3). Abstraction is a mental act by which we isolate a particular meaning from a whole set of perceptions and treat it separately, while other meanings are either completely abandoned or simply put aside. For example, if we keep only the number five

from the perception of five apples, we no longer think of apples but concentrate only on the number five (Stres 2018, s.v. "Abstrakcija"). We consider the resulting hypothesis to be the basis of an abstract principled rule, which is a major premise in the deductively valid argument *Modus ponens*:

> Major premise: p ⊃ q (abstract principled universal rule)
>
> Minor premise: p
>
> Conclusion: q

Example:

> Major premise: If we go to Lake Bled, we must come across a white swan there.
>
> Minor premise: We are at Lake Bled.
>
> Conclusion: We will certainly come across a white swan.

When we speak about rational intuitive abstractions, we usually mean the principles of formal logic like *The Principle of Sufficient Reason*, which states that everything must have its reason, or moral principles like the principle of the Golden rule, which says that we must treat other people fairly and with respect. We could claim, however, that we also have "*de re* or objectual intuitions of properties or states of affairs" and that "our *de re* grasp of various properties is what grounds or justifies our assent to propositions involving them" (Zalta 2019, s.v. "Intuition"). The principled rule we used for a major premise (If we go to Lake Bled, we must come across a white swan there.) is an example of the objectual intuition of a state of affairs, whereby the latter does not change but in principle remains the same.

The precedence of parts over the whole also manifests itself in the fact that there is no prior knowledge (a whole) needed for the discovery of the principles or abstract universal rules. We mentioned earlier that there is a tacit knowledge involved in the production of principles, for the more we are experienced the more efficiently we produce right principles. However, tacit knowledge, unlike theoretical knowledge, is not holistic—the latter can be described with the "web" metaphor—but is particular and limited to a specific area. The tacit knowledge of using a mobile phone cannot, for example, help one to find correct moral principles.

(B) In the history of modern science, the deductive precedence of parts over the whole manifests itself in the beginnings of modern science, when scientists began to observe the state of particular natural phenomena with the help of different scientific instruments and to record observations, in which they intuitively grasped meaningful abstract patterns that could be described with mathematical equations of scientific laws. One of the paradigmatic examples of this new science was the discovery of Kepler's First Law or principle, which states that planets move around the Sun in an elliptical orbit, which can be described mathematically. Ancient philosophers believed that the planet's orbit was circular, but they did not verify their belief with observations. The elliptical orbit of a planet was intuitively grasped by Johannes Kepler, who studied astronomical observations of the planet Mars. Mars is in the state of orbiting around the Sun, which is the same state as that of the planet Jupiter, which is also orbiting around the Sun. Description of the state of one planet is valid also for the other planet because they are in the same state. Before Kepler, Galileo Galilei discovered, with the telescope he himself invented, that celestial bodies such as the Moon are not perfect, as ancient wisdom claimed, but imperfect, for the surface of the Moon turned out to be not smooth but mountainous. From the observations of perceived attributes of natural phenomena, these scientific pioneers intuitively grasped the right abstract principled picture of the world, which the ancient sages had misunderstood. The most famous example of a modern scientist is, of course, Nicolaus Copernicus, who intuitively replaced the ancient geocentric picture of the world with the heliocentric model on the basis of astronomic observations of particular movements or situations.

In science the deductive precedence of parts over the whole manifests itself also in the fact, that for the discovery of a scientific law no prior knowledge (a whole) is needed (e.g., for the discovery of elliptical orbit). A scientist needs just observations and deductive rules with the help of which a scientific law is deduced. In addition, a scientific law or mathematical theorem can be proved by an individual scientist without the help of the scientific community.

(C) In society, the deductive precedence of parts over the whole manifests itself in the autocracy, which was the system of government in the ancient Roman Empire and in the medieval Byzantine Empire, as well as in the French, Spanish, Prussian and Austrian absolute monarchies and in Nazi Germany. In autocracy all power is concentrated in one particular person, namely the autocrat who has absolute power over all other parts, i.e., the whole, which is nothing more than the sum of individuals. The autocrat's subjects are not connected into any social groups but are atomic units united by force and loosely interconnected. The Communists, for example, forbade individuals from grouping into individual social groups. An autocrat does not consider his subjects as persons but only as material components that constitute society. They have no rights and can be removed from the whole, if necessary, without the whole suffering or being impaired as a result. An individual is dispensable and replaceable with another, for there is always someone who can occupy the vacant position. Stalin, for example, had no problem sacrificing millions of his subjects in the fight against Hitler because they were only consumables for him. The individuals have different values, and society has a pyramid structure (Figure 3), with the autocrat and his aristocracy at the top. They have the highest value because of their supposed perfection, in contrast to the multitude of supposedly imperfect and ignorant individuals. The autocratic society is a classic example of the Great Chain of Being, first put forward by ancient Greek philosophers.

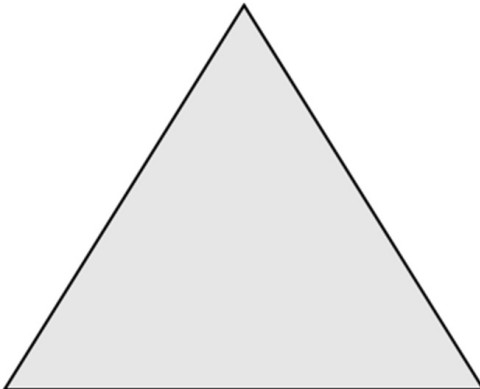

**Figure 3.** The social pyramid of the autocratic society.

3.1.2. The Highest and the Lowest Position: The Supremacy of Abstract Rules over Concrete Observational Facts

In the material world, an abstract principled rule is at the highest position and has the greatest importance, while a concrete individual fact has a lower value. The abstract has greater importance than the concrete because, with the increase in abstraction, an increase in perfection is also supposed to happen. Human knowledge is strictly hierarchical and has the shape of a pyramid, with the most valuable abstract disciplines, such as physics and mathematics, at the top.

(A) In logic, the deductive supremacy of abstract rules over concrete observational facts is reflected in the *Modus ponens* basic schema (argument) of reasoning, which is also a method of justification and is included in practically every formal logical system (Encyclopædia Britannica 2007. S.v. "Modus Ponens and Modus Tollens"). This supremacy is also reflected in the fact that a concrete fact (a thesis) is proven by an abstract rule (and not exactly the other way around) and that counter-evidence cannot refute the abstract universal rule because the conclusion of an argument is not "necessarily" true but just

"certainly" true, for the correlation is not causal but just a regularity. The reasoning schema of the argument is as follows:

> If we go to Lake Bled, we must come across a white swan there p $\supset$ q (abstract rule)
>
> We are at Lake Bled. p
>
> We will certainly come across a white swan. q

Based on this schema of reasoning (argument), we can justify the following statement (thesis): We will certainly come across a white swan, for we are at Lake Bled, and the abstract principled rule states that if we go to Lake Bled, we must come across a white swan there. We can clearly see that in this case, the concrete fact (a thesis) is proved by an abstract rule and not vice versa.

If we do not encounter a swan at Lake Bled, however, this does not mean that the abstract principled rule is overturned because the conclusion is by definition only certainly true, but not necessarily true. Namely the correlation between visiting Lake Bled and encountering a swan there is not a causal one but is only a regularity. If we go to Lake Bled, we must come across a white swan there in principle, but this, however, does not necessarily have to happen for the abstract rule to remain true. This means, however, that an abstract principled rule cannot be refuted by counter-evidence. Thus, the deductive supremacy of abstract rules over concrete observational facts is reflected in the inviolability of the abstract principled rule, which in the deductive argument is in the highest, most sacred position, above the concrete facts.

(B) In science, the deductive supremacy of abstract rules over concrete observational facts is reflected in the classical hierarchical classification of scientific knowledge, in which the most abstract disciplines, such as mathematics and logic, are at the top because they are the most important. It is also reflected in the use of the deductive *Modus ponens* argument for making predictions of natural phenomena or proving concrete observational facts by the abstract principled rules.

> Example: Justification of the thesis that planet Mars will orbit the Sun in an ellipse.
>
> Premise 1: Kepler's First Law (or better, principled rule) states that if a planet moves around the Sun, its orbit must have the shape of an ellipse, with the Sun located in one of the foci of the ellipse.
>
> Premise 2: Planet Mars orbits the Sun.
>
> Conclusion: The orbit of the planet Mars will certainly have the shape of an ellipse.

Based on the *Modus ponens* inference schema, we can justify the following thesis: The orbit of the planet Mars will certainly have the shape of an ellipse because the planet Mars orbits the Sun, and Kepler's First Law states that if a planet orbits the Sun, then its orbit must have the shape of an ellipse, which can be described by a scientific law or equation $E$.

A single counter-evidence cannot refute a scientific law. For example, if the measurements do not agree with the predictions made by a scientific law, scientists usually do not reject the law, but look for a theoretical explanation that explains the anomaly. The reason for the deviation from the predicted movement of the planet may be, for example, the gravitational influence of a neighboring planet.

(C) In society, the deductive supremacy of abstract rules over concrete observational facts is reflected in the fact that in an autocratic society, an individual has no influence on the making of the abstract rules, to which subjects must submit because this is exclusively in the domain of the autocrat. Furthermore, an individual is completely subordinate to these rules, which are considered inviolable. An individual is also without any rights, and his opposition to the rules leads to his elimination by the autocrat. The abstract rules of society are completely inviolable, and the individual is absolutely subject to them, similarly to the rules of science and logic.

Based on concrete examples from logic, science and society, we have shown that the initial state of Western society is characterized by a deductive pattern of thinking, because the part takes precedence over the whole, and the concrete is subordinate to the abstract.

### 3.2. The Second, Inductive Phase of Western Society

The inductive pattern of thinking which has emerged during the early modern period stems from the lived world or lifeworld, which is the flesh of the material world. The lifeworld or the flesh of the material world is the source of our understanding of reality because by discovering the essential properties of natural phenomena, the probability of deepening our understanding by discovering a meaningful explanation increases. We belong to the lifeworld with the soul, and we perceive its meaning on the basis of lived experience and grasp it with techniques of inductive generalization. There is an antisymmetric relation between the material world and the lifeworld with respect to the highest and the lowest position.

A convenient symbol of inductive logic is England, which is the birthplace of inductivism, human rights and liberal democracy. These cultural innovations are, in our view, an expression of the inductive pattern of thinking, which—as in the case of deductive logic—gives priority to parts over the whole and—similar to the Gospel's logic—considers the abstract to be subordinate to the concrete. We will explain what this means in concrete terms with the following three concrete examples. We will also show, however, that Western society, by subordinating the abstract to the concrete, has taken a step closer to Gospel's logic of double antisymmetry.

### 3.2.1. The First and the Last Place: The Precedence of Parts (as Building Blocks) over the Whole (as a Framework, Structure)

The first and the last place have the same line-up as in the material world; namely, the part takes precedence over the whole, which means that the part remains more important than the whole, and the whole is still nothing more than the sum of the individual parts. However, in the lifeworld, we are not talking about oneness in the multitude but about the unified framework (structure, building) built from a multiplicity of building blocks. Each building block is an essential element of the framework, but the whole can exist despite the absence of a building block, which means that they are not indispensable. For example, a skeleton remains a skeleton despite the fact that a bone has been removed from it; the bone also exists separately from the skeleton. The framework of a liberal democratic society, for example, is built from the interaction of the following building blocks: interest organizations, political parties, the civil service system, the media, and the atomized mass of people. Even if any of these essential parts of democracy are missing (when, for example, an interest organization or political party ceases to exist), democracy can still function, only in a slightly truncated form, but we cannot infer from the properties of the building blocks the property of the whole framework. For example, if one of the political parties is anarchic, it does not mean that a democratic society, as such, is anarchic.

(A) In logic, the inductive precedence of parts over the whole manifests itself first in the inductive methods of discovery, namely in the inductive generalization, which is composed of different inductive techniques of generalization (e.g., Mill's inductive methods). With their help and based on a multitude of particular experiences of an event (e.g., the events of adaptation), we obtain a general law or an outline of the quiddity of this event of a natural phenomenon (e.g., the general law of adaptation). As in the case of deductive abstraction, also in the case of inductive generalization, the precedence of parts over the whole is manifested in the fact that prior theoretical knowledge (a whole) for the discovery of quiddities is not needed because this arises from the generalization of a multiplicity of particular atomic experiential facts.

(B) In science, the inductive precedence of parts over the whole manifests itself for example in the discovery and in the use of inductive (Hume's) generalization techniques of discovery, which do not need any prior knowledge. Inductive generalization was the source of discoveries of naturalistic general laws since the beginning of the Scientific Revolution in the 16th century Europe, which culminated in late 17th century with the founding of the Royal Society and the appearance of Isaac Newton, who was its most famous member. Newton discovered, among other things, three Laws of Motions. Darwin discovered the general Law of Evolution, which says that living beings adapt to their environment. Hume's generalization, which is both a method of discovery and a method of justification, has the following inference schema:

For all hitherto known elements x of the set M, the statement Fx is true.

For all elements x of the set M, the statement Fx is true.

Example: Discovering the inductive general law of conductivity of the electric current using the experimental inductive generalization technique.

Premise 1: in experiment no. 1, copper wire conducts the electric current.

Premise 2: in experiment no. 2, copper wire conducts the electric current.

. . .

Premise n: in experiment no. n copper wire conducts the electric current.

Conclusion: the copper wire probably conducts electric current.

(C) In society, the inductive precedence of parts over the whole manifests itself in the liberal democratic social order. Liberal democracy as an idea together with the idea of political parties was first expressed in England in the 17th century. Social order is built of building blocks that include interest organizations, political parties, the civil service system, the media, and the atomized mass of people (Hansen 2004, p. 12). In the case of interest organizations, we can speak about civil society (i.e., nongovernmental organizations) that implements the public interest (nonprofit organizations, cultural and religious groups, humanitarian organizations, online groups, social media communities, innovators, media, etc.).

All of these building blocks build the structure of a democratic society, with the whole still being just the sum of the individual parts, the same as in an autocratic society, because there is no organic connection between the parts of the whole. The building blocks have different levels of importance and power, which means the democratic society is hierarchically structured in the form of an upright social pyramid, in which the atomized mass of people is the least important and has the least power. The power is concentrated in individual organizations and political parties, and the individual shares in this power, insofar as it belongs to them. Thus, unlike an autocracy, people in a democracy are connected to certain social groups, and these groups, as building blocks, build the entire framework of society. However, despite being part of a particular social building block (for example, being part of a political party), an individual is still without a personal identity, and his or her individual opinion is disregarded if over-ruled by a majority. His vote has value only as much as it is part of the majority. As an individual, however, a person is powerless and cannot influence the decisions made by society unless he joins some group that demonstrates publicly in front of parliament. A social whole can, therefore, exist and function without a particular social group or an individual, whose presence is not indispensable, and an individual and a social group can also exist and function without a democratic society.

In a democratic society, just as in an autocratic society, people are not organically connected to each other, so they do not rejoice with those who rejoice and do not weep with those who weep. Individual groups of democratic society compete with each other and are sometimes even in mutual hatred, such as political parties and political poles, when the democratic community seems to be divided into two poles, and each pole wants to harm the other. In international competitions, we do not have representatives of the country's



democratic society but state representatives. The connection between individuals is just superficial, but is still stronger than in the case of autocratic society, because people are connected to certain social groups. In this case we can see, that inductive precedence of parts over the whole is more progressive than the deductive precedence.

### 3.2.2. The Highest and the Lowest Position: The Supremacy of Concrete Individual Experiential Facts over Abstract General Laws

There is a vertical antisymmetric relation between the lifeworld and the material world regarding the evaluation of importance of the abstract and concrete because in the lifeworld, concrete individual experiential facts are more important and are above abstract general law (hypothesis), which is just the opposite as in the material world and, as we shall see, similar to the Gospel's personal life. The importance of an abstract principled rule in the material world is the same as the importance of a concrete individual experience (fact) in the lifeworld, which is the basis for mutual antisymmetry because the importance stays invariant; just the poles are switched. What is important for deductive logic is unimportant for inductive logic and vice versa.

(A) In logic, the inductive supremacy of individual experiential fact over general law is reflected in the basic inductive schema of reasoning (argument), called Hume's generalization, which is "the most important type of inductive reasoning, without which no general laws could be reached in the experiential sciences." (Uršič and Markič 2009, p. 264). It is also reflected in the fact that the abstract general law is proven by the concrete experiential facts, which is different than in deduction, where the concrete is proven by the abstract, which is why we can speak about vertical antisymmetry between the two. The example of proving the inductive general law of the breaking of a wooden board is as follows:

Premise 1: The wooden board 1 broke under load.

Premise 2: The wooden board 2 broke under load.

Premise 3: The wooden board 3 broke under load.

. . .

Premise n: The wooden board n broke under load.

Conclusion: A wooden board under (an excessive) load will probably break.

Based on this schema, we can form the following argument: The wooden board under (an excessive) load will probably break, for the board always broke in all load experiments.

In this case of a wooden board we can see that inductive logic is in vertical antisymmetric relation with deductive logic because the abstract general hypothesis of an inductive argument is now at the bottom of an argument, and the concrete individual factual premises are above it. This is certainly not just an aesthetic move but is manifested in a completely different way of thinking because inductive generalizations are not sacred and inviolable as deductive principled rules are because a single negative experience has the authority to refute an abstract general hypothesis. For example, if one wooden board did not break in an experiment, the general hypothesis that a wooden board under (an excessive) load will probably break would be refuted, no matter how many cases of board breaking we have. This problem is known in philosophy as the problem of induction, pointed out by the Scottish philosopher David Hume and more recently by the philosopher of science Karl Popper. The problem of induction is that observations can repeatedly confirm a particular hypothesis, and there is still a theoretical possibility that the next measurement will refute the hypothesis (Bem and de Jong 2006, p. 72).

(B) In science, the inductive value supremacy of concrete individual experiential fact over abstract general law is reflected in the use of inductive hypothesis justification techniques, which are based on Hume's generalization method, in which an individual experiential fact has the power to refute a general hypothesis. We can clearly see this in the case of the widely used method of the inference to the best explanation, which can be written as follows:

Premise 1: It is reasonable to believe that the best explanation of the facts available to us is also true.

Premise 2: *D* is a fact.

Premise 3: Hypothesis *H* explains *D*.

Premise 4: None of the competitive hypotheses available to us explains *D* as well as *H*.

Conclusion:Thus, it is reasonable to believe that hypothesis *H* is true.

In this case, we can clearly see how, based on a few examples of less competitive hypotheses, we draw the general conclusion that our hypothesis *H* is the best explanation of all possible explanations. However, even a single better hypothesis *H′*, which better explains fact *D* than *H*, refutes the inductive hypothetical conclusion that it is reasonable to believe that hypothesis *H* is true.

(C) In society, the inductive value supremacy of concrete individual experiential fact over abstract general law is reflected in the fact that the individual, as a building block of the society, has civil rights, which are inalienable; furthermore, in court, a general law that deprives him of these rights can be annulled. A civil right of the individual is so sacred that it is inviolable and superior in value to general state law. In the event that systemic human rights violations are found in an EU Member State, the European Court of Human Rights may also impose an obligation on the state to adopt a legal basis in the domestic legal order. However, because of the primacy of parts over the whole, individual organizations only pursue the shared interests of their members, including their rights, and people who do not belong to a strong organization have difficulty defending their civil rights. Civil rights (liberties) in Great Britain have begun as early as with the Magna Carta of 1215, which is the written precursor to many of contemporary human rights documents. In a democratic society, there is no deeper organic connection between individuals and different organizations look solely after themselves, but we can clearly see that a democratic society is a great advance compared to an autocracy.

Based on these examples from logic, science and society, we can clearly see, that the supremacy of the concrete over the abstract brought the progress to Western society. This pattern of thinking is also a characteristic of the Gospel's personal life, which means that by moving to an inductive pattern of thinking, Western society has taken the first step towards its conversion. However, it did not stop there, but was already preparing to take a step towards an abductive pattern of thinking.

### 3.3. The Third, Abductive Phase of Western Society

The abductive pattern of thinking, which has emerged during the late modern period, stems from the material life. Material life is intelligible, meaningful patterns of reality happening. The evolutionary struggle for survival, for example, is material life's pattern of a happening that, once we perceive it, allows us to comprehend the natural phenomenon of evolution. We belong to material life with our heart, we perceive its meaning with our senses, and we comprehend it with abductive logic. The philosopher Martin Heidegger spoke of the forgetfulness of Being, but we can speak of an even greater forgetfulness, namely, the forgetfulness of the reality of life, which is taken as something derived from the material world, even though it is not of that world despite being in the midst of it.

Material life is in horizontal antisymmetric relation to the material world and in double-antisymmetric relation to the lifeworld, which means the combination of horizontal and vertical antisymmetry. The antisymmetry is based on the fact that the importance of the part (A) for the material world is the same as the importance of the whole (B) for the material life, i.e., stays invariant so that there exists the antisymmetric relationship: AB = BA. Material life is also vertically antisymmetric to the lifeworld because its abstract meaningful life guides are more important than concrete individual experiential facts, while in the lifeworld, concrete individual experiential facts are more important than general laws.

The material life is reflected in the abductive pattern of thinking, in which the whole takes precedence over parts, but the concrete is, similar to the material world, subordinate to the abstract A convenient symbol of abductive logic is United States of America because it is the birthplace of abductivism and of the first Constitution and is renowned for famous theoretical scientists. A new step in the direction of the conversion of Western society, which is in fact its deepening, is in giving priority to the whole over the parts, as it enables the organic connection of the parts with each other.

### 3.3.1. The First and the Last Place: The Precedence of the Whole (as an Organism) over the Parts (as Segments)

In material life, the whole takes precedence over the parts, with the whole being the organism and the part its individual segment. We are talking about a plurality of segments that are united in a single organism, which means that there is a plurality in unity. The whole, in this case, is more than just the sum of the individual parts; this addition is an impersonal identity (cf. Michelini 2020).

We can observe that material life is horizontally antisymmetrical to the material world because the first and last places are interchanged, which means that, in the first place, there is no longer a part but a whole; thus, it is not oneness in the multitude, but the plurality in unity. In material life, the whole is more important than its parts. The whole is a living, whole organism, broken down into many segments, where neither the segments nor the whole can exist on their own without reference to another.

If we take an animal cell, for example, we can find that it is made up of a plurality of organelles such as membrane, nucleus, mitochondria, proteins, DNA, and similar elements; despite the plurality of segments, it is a single organism that has an impersonal identity. If we isolated an individual segment and tore it away from the whole, both the individual segment and the whole would cease to exist. This is especially evident in diseases when one organ fails, and, as a result, the whole organism collapses. This is just the opposite to in a democratic society, where the absence or a collapse of a particular organization does not affect society as a whole.

An individual segment is thus still indispensable and valuable in an organism that bears an impersonal identity, but it is usually found in several copies, which are, however, dispensable as individual units. An indispensable cellular organelle, for example, is the mitochondrion, which supplies the cell with energy. For example, there are many mitochondria in a cell; if one fails, hundreds of others are still available. We can see that parts are important and indispensable only as particular organs, while individual representatives or copies of this organ are still not indispensable, which means that importance and power has not yet reached the lowest part of the hierarchical pyramid of being.

In material life, we can infer from the characteristics of individual parts to the characteristics of the whole because the segments and their organism are inextricably linked. From the beauty of individual parts, for example, we can infer the beauty of the whole. The individual segments of the organism are hierarchically organized in the form of a pyramid, with some having greater value or importance than others.

(A) In logic, the abductive precedence of a whole over its parts manifests itself in the method of discovery, abductive reasoning, in which, unlike in the deductive and inductive methods of discovery, we do not start with individual parts but with the whole of reality, namely with the prior knowledge we have about the world and life. In this totality of knowledge, upon contact with surprising individual facts, a meaningful explanation is abductively born, giving meaning to those facts and connecting them into a meaningful whole. Charles S. Peirce formally wrote the abductive method of making sense of surprising phenomena as follows (except premise 1, which was added by the author):

Premise 1: We possess preexistent knowledge about the world and life

Premise 2: The surprising fact, C, is observed

Premise 3: But if A were true, C would be a matter of course.

Conclusion: Hence, there is reason to suspect that A is true.

(Peirce 1960)

There is no technique for discovering meaningful explanations because these only come at the moment we least expect, even while sleeping. McGrath says this: "Sometimes hypotheses arise from a long period of reflection on observation; sometimes they come about in a flash of inspiration" (McGrath 2010, p. 95). Insight, which is crucial for the creation of an explanation is called differently by different authors: good feeling (Pierre Duhem), pragmatic guessing (Willard Quine) or guessing instinct (Charles Peirce). Explanation gives meaning to surprising phenomena so that they are not ignored but made meaningful.

(B) In science, the abductive precedence of a whole over its parts manifests itself in the use of the method of abductive reasoning, which is a method of discovery, and in the history of science in the establishing of scientific explanations on the scientific scene in the 19th century, when theoretical phenomena such as the evolutionary struggle for survival and the electromagnetic field and atoms were discovered. Abduction as a form of inference was first put forward by an American philosopher and scientist Charles Sanders Peirce (1839–1914) in 1901. However, the public initially nurtured considerable distrust of scientific explanations because these describe unobservable processes that cannot be empirically proven. This was the case until the second half of the 20th century, when the public finally began to trust and value them as was their due. This, in turn, has led to the fact that today, the scientific explanation is the most valued product of science, and the public and scientific community are eagerly awaiting the discovery of the "theory of everything". The United States in particular is famous for renowned theorists such as theoretical physicist Albert Einstein (General theory of relativity), molecular biologist James Watson (DNA discovery), population genetics Sewall Wright (neo-Darwinism) and Ralph A. Alpher (Big Bang theory).

In science, the abductive precedence of a whole over its parts also manifests itself in the structure of scientific explanation itself, which consists of individual hypotheses that are connected into a meaningful whole, and every hypothesis is further connected to many auxiliary hypotheses. With this web of hypotheses, however, the phenomenon of confirmation holism and holistic underdetermination is associated. When it turns out that a scientific explanation makes a wrong prediction or is contradicted by evidence, due to the holistic underdetermination, we cannot test its hypotheses individually. This is because hypotheses have empirical consequences only in conjunction with other hypotheses and/or auxiliary hypotheses about the world. "When the world does not live up to our theory-grounded expectations, we must give up something, but because no hypothesis is ever tested in isolation, no experiment ever tells us precisely which belief it is that we must revise or give up as mistaken" (Zalta 2016, s.v. "Underdetermination of Scientific Theory").

(C) In society, the abductive precedence of a whole over its parts manifests itself in national unity and in a united state, which is more than just the sum of individual parts, for it is the bearer of impersonal identity, which is reflected in the fact that the state is often portrayed as a human figure (e.g., Germany is portrayed as the figure Germania). A democratic society does not have its own identity; for example, we never talk about a Slovene democratic society but about the Slovene nation or the Slovene national state. A state, which can be a republic or a monarchy, has a constitution that sets out guidelines for the common life of citizens as a whole, which means that the whole is before the individual part, the constitution before the law. The Constitution of the United States, which was ratified in 1788 is often cited as the oldest Constitution in the world. The Constitution and the constitutional state are signs of the progress of a society that has become a nation with its own piece of land.

The constitutional state is an organism that is broken down into individual parts. The state is represented by united state bodies such as the president (or king, prince, emperor, oligarch), the state council, the state administration, the constitutional court, the constitution and other state bodies and legal entities under public law (agencies, public funds, social

security providers), which have different values and are hierarchically organized in the form of a hierarchical pyramid turned upside down. Without individual state bodies, the state cannot exist. For example, if the courts were abolished, the state would collapse. The hierarchy of the society is horizontally antisymmetric to autocracy and democracy because individual state organs are supposed to serve its citizens, which is indicated in the names like "minister", which has its origin in Latin and means "servant". Such an upside-down social pyramid, in which a president of a state is located at its lowest tip and the citizens are located at its top, is certainly a radical step in the conversion of Western society, as the pattern of thinking is turned upside down (Figure 4).

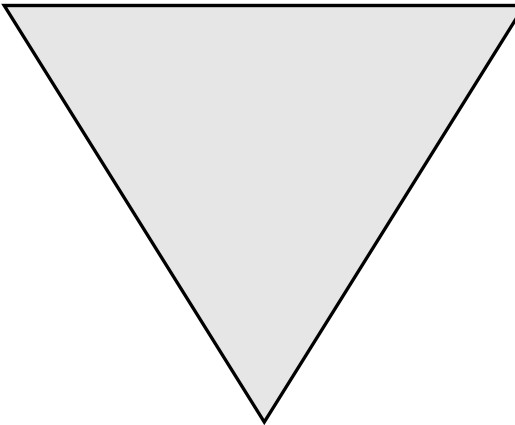

**Figure 4.** The upside-down social pyramid of the constitutional state.

Citizens feel connected to the body as segments of that body, and if one part suffers, everyone else suffers. Furthermore, if one part rejoices, all the other parts rejoice. This is shown, for example, at sporting events, when the joy of the winning sports team of the national team delights all citizens. Moreover, when a company collapses and people lose their jobs, the pain affects everyone who lives with them in the same state. This is completely different than in the case of a deductive autocratic or inductive democratic society, in which individual groups of individuals compete with each other and even try to harm each other. Here we can clearly see the antisymmetric relation and a clear sign that a society has converted towards the Gospel's universal brotherhood.

In a constitutional state, power is not concentrated in a single person as in autocracy or a particular organization or political party as in democracy; it is usually separated in three organs—Legislature, Executive and Judiciary—which have a duty to serve the public. The individual citizen is an important part of the state and has a certain amount of power, especially by having a voice in the election of the government (parliamentary elections, presidential elections). The precedence of a whole over its parts is also reflected in political federations, such as the United States, and in international alliances, such as the European Union. In all these political bodies, unanimous decisions are taken by voting, which means that a simple or qualified majority must agree to take a decision. The disadvantage of unity in the plurality of votes is that this unity is not perfect because no absolute majority is required, only a simple or qualified majority. An individual citizen does not have his or her own identity because only individual state bodies have it, and his or her vote is only one vote out of many voices that others can easily overpower. An individual citizen is only one copy of a particular state organ, e.g., the public, and is easily overlooked, outvoted or replaced with another copy. Thus, although the hierarchical structure of the constitutional state is antisymmetric to autocracy and democracy, the individual still lacks the importance and dignity for which his or her voice would necessarily be taken into account. The conversion of the society is still not complete.

### 3.3.2. The Highest and the Lowest Position: The Supremacy of Abstract Meaningful Life's Guides over Concrete Individual Facts

In material life, as in the material world, an abstract meaningful life's guide is at the highest position and has the greatest value, while a concrete individual fact has lower importance and is placed in a lower position. The plurality of concrete facts is subordinate to the abstract summit in both cases, which means that there is no antisymmetry between the material world and life but just symmetry. Nevertheless, there is clear antisymmetry between material life and the lifeworld because in the latter, as we saw earlier, concrete individual facts have greater authority than abstract general laws.

(A) In logic, the abductive supremacy of abstract meaningful life's guides over individual life's experiential facts is reflected in the basic schema of inference (argument) *Affirming the consequent*, which is a deductive invalid variant of the schema of inference *Modus ponens*, and is a symmetrical inversion of deduction (Uršič and Markič 2009, p. 260), in which the first and the last place are replaced (i.e., the conclusion and the effect). It is about horizontal antisymmetry. *Affirming the consequent* has the following schema:

Premise 1: If it rains, the roads can be wet. $p \supset q$ (If hypothesis $H$ holds, then fact $D$ can occur)

Premise 2: The roads are wet. $q$ (The fact $D$ occurs)

Conclusion: It is possible that it is raining. $p$ (It is possible that hypothesis $H$ holds)

In life, there are many possibilities open, which is why the general guide says that the roads *can* be wet. Based on this reasoning schema, we can justify the following assertion (thesis): It is possible that it is raining because the roads are wet, and if it is raining, the roads can be wet.

Symmetric with the principled rule, the meaningful life's guide is also sacred or persistent and cannot be refuted by one counter-evidence because the form of the argument does not allow it and because the general guide only says that the fact $D$ can occur, which means that it is possible that it will not occur.

In our case, therefore, the fact that the roads are not wet does not invalidate the abstract meaningful life's guide because the absence of rain as the reason for the dryness of the roads is only one of the possibilities. This, however, is a perfectly logical conclusion because we can conclude that in the context of life, there is a possibility the roads are not wet, for example, because someone laid a tarpaulin on the road. The above general meaningful life's guide remains valid and meaningful even though we have found contrary evidence for it, which means that it is by nature sacred, inviolable, and enduring, as is the abstract principled worldly rule of the material world. In this case we can see clearly the symmetry between the two, and both are in antisymmetry with lifeworld.

(B) In science, the abductive supremacy of abstract meaningful life's guides over concrete individual life's experiential facts is reflected in the use of *Affirming the consequent* to confirm scientific theoretical explanations, where the concrete experiential facts are proven by the abstract life's guide, as in the case of the theory of photosynthesis:

Premise 1: If a plant produces glucose and oxygen from $CO_2$ and water with the process of photosynthesis (an explanation), then, as a result, it can gain most of its weight from the air and can grow on bare rocks. (An abstract meaningful life's guide)

Premise 2: We see the individual tree growing on a bare rock. (a concrete experiential fact)

Conclusion: It is possible that a tree produces glucose and oxygen from $CO_2$ and $H_2O$ with the process of photosynthesis.

The justification is as follows: It is possible that a tree produces food from $CO_2$ and $H_2O$ with the process of photosynthesis because in hills, we often see trees growing on a bare rock, and an abstract meaningful life's guide says that if a plant produces food from

$CO_2$ and $H_2O$ with the process of photosynthesis, consequently it can gain its weight from the air and can therefore grow on bare rock.

However, the absence of trees growing on bare rock does not mean the falsification of the abstract life's guide and its theoretical explanation because this possibility is still given and can be exploited by a tree at any time in the future.

The fact that there is only a "possibility" and not a "necessity" of a theory (meaningful explanation) can be also explained by the thesis of contrastive underdetermination, which says that "any body of evidence confirming a theory, there might well be other theories that are also well confirmed by that very same body of evidence." (Zalta 2016, s.v. "Underdetermination of Scientific Theory"). In our case, we could say that there is a possibility that the observance that individual trees can grow on bare rocks is also evidence for some competing theory that has a different meaning. This is the so-called empirical equivalence thesis. The Duhem–Quine thesis similarly states that "all theories entail observational consequences only with the help of auxiliary assumptions", and that "for any evidence and any two rival theories T and T', there are suitable auxiliaries such that T' and suitable auxiliary assumptions will be empirically equivalent to T together with its own auxiliaries." (Psillos 1999, p. 158).

(C) In society, the abductive supremacy of abstract meaningful life's guides over an individual life's experiential facts is reflected in the fact that the individual citizen, despite participating in state affairs through elections, does not have such authority to overturn a constitutional amendment because this is the jurisdiction of the elected government. As a citizen, he has no identity and only his voice counts, but it is easily ignored or neglected. Just as one piece of counter-evidence cannot disprove a theory, an individual citizen, despite sound arguments, cannot influence the adoption of amendments to the constitution in parliament. Moreover, this applies not only to individual, but also to social groups, such as different Christian Churches, which, despite sound arguments, cannot achieve the abolition of the constitutional right to abortion. This lack of authority of the individual citizen before the state is in stark contrast with the authority of an individual before the democratic society, who can, with the help of a referendum or a court, achieve the abolition of a law that violates his civil rights.

In the case of the constitutional state, we can see that society has not made progress in terms of the supremacy of the concrete over the abstract, but only in terms of giving priority to the whole over parts. The constitutional state still does not treat the individual as a person in the Gospel's sense, which means that society still needs the last step of conversion, namely to become a society in which its members have a face or true identity.

### 3.4. The Final, Eductive Phase of Western Society

The eductive pattern of thinking, which has emerged with the advent of the Postmodern era in the second half of the 20th century, stems from the personal life, which is the origin of our understanding of reality. We belong to the personal life with our spirit, and we perceive the personal original and personal purposive meaning by personal experience and further comprehend it with the help of eductive identification and arguments. The eductive logic we use here has no justification in the literature, so we propose it here only as a theoretical proposition based on common sense. Just like material life, personal life as an independent reality is a neglected topic that still needs theoretical treatment.

Personal lives are comprehensible original and purposive models of reality identification. Reality of life not only happens but also identifies itself through people, with the identification process inside people being more dynamic than the external happening of material life itself. Furthermore, from the identification with a certain origin (e.g., a doctor) and purpose, a personal original or purposive identity is born, which is the revealed truth about a person. Someone identifies himself with a mother, father, doctor, soldier, priest, athlete, electrician, and so on. From this, his maternal, paternal, military, or some other identity is born. All of these original and purposive meanings are models, which means they already exist, buried in the deepest part of reality; we just have to discover them. We

need to comprehend who is the true mother, father, soldier, etc. The role model of the mother is, for example, the Virgin Mary, who is the model of immaculate maternal love. Mary identifies herself with the Mother of God, which means that the truth about her is the identity of the Mother of God.The personal life is reflected in the eductive pattern of thinking, in which the whole takes precedence over parts, and the abstract is subordinate to the concrete, which means that this is the last step in conversion towards the Gospel's double antisymmetry with respect to the material world. The personal life is characterized by horizontal antisymmetry with respect to the lifeworld (in personal life, the whole is before its parts), by vertical antisymmetry with respect to the material life (in personal life, concrete individual personal experiential facts are above abstract original advice or recommendation) and by double-antisymmetry with respect to the material world. The people of God of the messianic Kingdom of God are not an impersonal multitude as is true in the case of a democratic society, but a community of members concurred together with one another and have a personal identity of a mother, with the Virgin Mary being its image. In the same way that Mary gave birth to Jesus, the People of God gave birth to God's children through baptism.

Although, in our opinion, with the emergence of postmodern society, Western society has taken the last, decisive step towards its transformation into the People of God, there is still a long way to go before reaching complete conversion, because as a society we are still far from being one heart and soul, as was the case in the first Christian community (cf. Acts 4:32).

### 3.4.1. The First and the Last Place: The Precedence of the Whole (a Spiritual Body) over the Parts (as Members)

There is a symmetry between personal and material life in this case: in the former, as in the latter, the whole takes precedence over individual parts. Furthermore, in personal life, the whole is the spiritual (living, subjective) body, and its parts are individual members. We can speak about a multitude of members united in one spiritual body, that is, about a spiritual unity in a multitude of members. The whole, in this case, is more than just the sum of the individual parts, and this addition is a personal identity. An individual member is indispensable, and its absence leads to the non-existence of the whole. For example, we cannot imagine a true father without some of his original qualities, such as giving firm support to the family. An individual member cannot exist on its own if it is separated from the spiritual body. For example, an original characteristic (like a father gives support to his family) as a member cannot exist on its own without an origin, namely a father, which is a spiritual body. In personal life, we can easily infer from the properties of an individual member to the property of the whole (i.e., the body). We can infer, for example, from the characteristics of the individual member of the People of God to the characteristics of the People of God as a whole because the members and the spiritual body are deeply connected.

There is antisymmetry between personal life and lifeworld or material world because we can again observe the familiar antisymmetric pattern AB = BA, where A is a whole and B is a part.

(A) In logic, the eductive precedence of a whole over its parts manifests itself in the eductive method of discovering the original identity (we educe that someone is a doctor, a priest, etc.), for we need prior knowledge to identify a person. This is also true for the generation of abductive explanations, but not for the general laws and scientific laws (regularities) for which no prior knowledge is needed for the discovery. Education allows us eductive identification when, based on our prior knowledge of who a certain origin is (e.g., who is a doctor) and based on the experienced individual characteristics of a person (e.g., we experience someone as a strict person), we infer on the original identity of the person. It is important to emphasize the fact that this identity is extracted or educed from the inside out, which is why we call this method eduction because we extract the truth about a person from himself, namely, who he is by his general original identity. The product of

eductive identification is thus the truth about a person, meaning his abstract original life's identities or the specific purposive life's identity, such as the identity of the Merciful Father or his Firstborn Son, who are two examples of life's ideals who attract us. The same as in abductive reasoning, where prior knowledge is required to form a meaningful explanation abductively for the eductive identification of a person, we need prior knowledge about original meaning (i.e., who a person is). We propose the following schema of reasoning on the example of educing the identity of a person based on our prior knowledge of who a father is:

> Premise 1: The prior knowledge as a general original truth of life: If someone is a father, then he knows how to be strict, has inner strength, is protective and gives support to his family.
>
> Premise (Personal experience) 1: Michael strives to be strict.
>
> Premise (Personal experience) 2: Michael strives to have inner strength.
>
> Premise (Personal experience) 3: Michael strives to be protective and to give support.
>
> Conclusion: Michael perhaps has the original identity of a true father.

This is in horizontal antisymmetry with the deductive and inductive logic, where no prior knowledge is needed for discovering abstract universal rules and general laws.

(B) In science, the eductive precedence of a whole over its parts manifests itself in the humanities, which attempt to extract the truth about human beings, namely, who a human being is. Wilhelm Dilthey saw understanding as the key for the human sciences, in contrast with the natural sciences. This precedence of a whole is also reflected in fact, that for the discovery of truths about humanity we need a prior knowledge, because we cannot analyze an individual identity in isolation from other identities. This is in horizontal antisymmetry with the deductive and inductive science, where no prior knowledge is needed for discovering a scientific law or a general law.

However, it is also reflected in the fact that the truthfulness of an individual scientific theoretical explanation is not recognized at the level of the individual, as scientific explanations cannot be proven, for the search for truth is in the domain of the scientific community. It is the whole of the scientific community that has the ability to judge the truth of an individual explanation. Moreover, this is the present situation of science through which the scientific community discerns the truthfulness of different competitive theoretical explanations. This is in horizontal antisymmetry with the deductive and inductive sciences, where an individual can for example mathematically prove a theorem or confirm a general law without the involvement of the scientific community.

(C) In society, the eductive precedence of a whole over its parts manifests itself in the eschatological Kingdom of God, in which the community of members is gathered in the united People of God, which has its original and purposive identity. The People of God has the original identity of a mother because just as a wife gives birth to children, so the People of God gives birth to children of God through baptism. Furthermore, the People of God has a purposive personal identity in the Virgin Mary, who is its purposive ideal image. As we have seen before, autocratic and democratic societies have no identity, and a state has only an impersonal identity. No member of the People of God is able to exist without the community, and neither is the People of God able to exist without an individual member. Each member has his or her personal original and purposive identity, and all members are gathered in the united People of God.

In the Kingdom of God, in contrast with the autocratic and democratic society, it is possible to infer from the characteristics of an individual member of the People of God to the characteristics of the whole because the members are befitting the whole and the whole befits them. Each member of the People of God has invaluable value and importance and is, as such, indispensable and irreplaceable.

Those who, in the eyes of the world occupy important positions, feel unworthy of that place and in their hearts consider themselves servants of the unimportant. The spiritual

hierarchical pyramid of the People of God is, the same as in constitutional state, a pyramid placed upside down, where the Holy Father is considered as the Servant of the Servants and is positioned on the lowest tip of the social pyramid. The Holy Father and other dignitaries are obliged to serve to the poor and all those who are worthless in the eyes of the world. The poor are a precious treasure for the Kingdom of God. The hierarchy of the People of God is thus horizontally antisymmetrical to the hierarchy of secular autocratic and democratic societies.

This utopian society of the People of God is in horizontal antisymmetry with the deductive and inductive society, where individuals are loosely connected with each other.

3.4.2. The Highest and the Lowest Position: The Supremacy of Concrete Individual Personal Experiential Facts over Abstract Original Advice and Abstract Purposeful Recommendations

In personal life as well as in the Kingdom of God, each personal experiential fact must be taken into account and, in the case of a single piece of counter-evidence, the hypothesis of the identity of a person must be discarded. This is in an antisymmetric relation with the material world and material life.

(A) In logic, the supremacy of concrete individual personal experiential facts over abstract original advice and abstract purposeful recommendations is reflected in the fact that abstract identity is proven by concrete individual personal experiential facts, which is symmetric to inductive logic and antisymmetric to deductive and abductive logic. The eductive scheme of reasoning is as follows:

Premise 1: Prior knowledge as an original truth of life.

Premise 2: Concrete individual experiential facts

Conclusion: this may be true original personal identity

It is clear from the schema that this is a holistic approach because identification takes place on the basis of prior knowledge of what an individual's original identity is. If we do not have this prior knowledge, then it is impossible to identify the person. On the basis of this schema, we can prove the thesis that a person has a certain original identity, as is evident in the aforementioned case:

Premise 1: The prior knowledge as an original truth of life: If someone is a father, then he knows how to be strict, has inner strength, is protective and gives support to his family.

Premise (Personal experience) 2: Michael strives to be strict.

Premise (Personal experience) 3: Michael strives to have inner strength.

Premise (Personal experience) 4: Michael strives to be protective and to give support.

Conclusion: Michael may have the original identity of a true father.

On the basis of the eductive schema of reasoning, we can justify the following statement (thesis): Maybe Michael is a true father because he strives to be strict, strives to have inner strength, strives to be protective and to give support, and the abstract original advice says that if you want to be a true father, then strive to be strict, to have inner strength, to be protective, and to give support to the family members.

The eductive conclusion that Michael is perhaps a true father can be confirmed by every new personal experience of him as a true father. Original identity is simultaneously the innermost and the deepest part of reality, which is the most difficult to reach or break through, as all three previous ways of perceiving meaning must be activated: observing the state of things, experiencing the event, and feeling what is happening.

In the evaluation of the true personal identity, a single individual experiential fact has the authority to refute an identity hypothesis, which is therefore not inviolable. We can show this very clearly in the case of the identity of the true father. For a person, we can make an eductive conclusion that he may be a father by his identity. However, when someone reports that he has experienced abuse at his hands, we refute this hypothesis and

begin to experience him as a criminal. This is in vertical antisymmetry with deductive and abductive logic, where a single counter-evidence cannot overturn an abstract universal rule or an abstract meaningful life's guide

(B) In science, the supremacy of individual personal experiential facts over abstract original advice and abstract purposeful recommendations is reflected in the authority of an individual to refute an accepted theory with a good argument. Scientific theories cannot be proven, but the scientific community gradually unites over time in the belief that a particular theory may be true, without ignoring any voice that has valid arguments for or against it. We know that a scientist can even win a Nobel Prize for obtaining counter-evidence for a theory, so refuting a theory is a great scientific achievement because it allows us to gain new perspectives on reality. In the scientific community, the authority argument does not apply, but only the authority of the argument. At present, we can observe that the scientific community has not yet come to be united in the case of many theories, but it cultivates the hope that this unity will only be achieved once in the future. The opposition of a single expert with a sound argument prevents the unity of the scientific community. This is in vertical antisymmetry with deductive and abductive science, where a single counter-evidence cannot overturn a scientific law or a scientific explanation.

(C) In society, the supremacy of individual personal experiential facts over abstract original advice and abstract purposeful recommendations is reflected in the eschatological Kingdom of God and its People of God, where not even one individual is overlooked because in the People of God, each member has his own identity and voice that is never ignored, provided that he has a sound argument. This eductive logic is, for example, manifested in the public character of the Catholic Church, as an individual member of the Church has the authority to refute one's identity as untrue on the basis of a sound argument. This is particularly evident in the case of the administration of the sacraments, which always have a public character, which means that even one well-founded concern against the administration of the sacrament to a certain person prevents it from being administered. For example, if two spouses want to be married, the public banns must be promulgated beforehand, and if only one person is found who has good reasons that the marriage cannot be established (e.g., that one of them is already Church-married), the priest must not and cannot administer the sacrament. If an individual submits a counter-evidence in the form of an experiential fact, he thereby refutes the identity of the individual as a candidate for receiving the sacraments and attributes to him the identity of a sinner who cannot receive the sacraments. This is in vertical antisymmetric relation with autocratic society and constitutional state, where an individual does not have the authority to challenge the autocrat's rules or constitutional amendments.

The eductive phase of Western society, in which, in our view, it entered with the advent of the postmodern era in the second half of the 20th century, is in a double antisymmetrical relationship with its initial, first phase. Only in the eductive phase both antisymmetries are present, both horizontal and vertical, which is a sign that Western society is well on its way to taking the final step of conversion and becoming the People of God.

Our four-layer reality model suggests that the Gospel's double antisymmetry is the final stage in the development of Western society, because according to the model there is no fifth layer, so the Gospel's double antisymmetry must be the end-point of the development of Western Society. Based on what has been said specifically about this double antisymmetry, we could say that no further step is needed, as the utopian society represented by the People of God is a life goal that cannot be surpassed.

The question is just how long it will take Western society to achieve this end-point and if it will be able to achieve it at all. Perhaps this life goal will forever remain only as an unfulfilled ideal of life.

## 4. Discussion

In this paper, we have pointed to the double antisymmetry that can be observed in analyzing all four areas of stratified reality, namely horizontal and vertical. We have

found that language itself takes into account the subtle differences that exist between these different layers; with the present explanation, we have attempted to theoretically process and order this wisdom of language as well, to make it intelligible. However, as our approach is fallibilistic and may have yielded erroneous results, our model will have to be tested with new concrete examples in the future.

The double-antisymmetry between the material world and personal life reveals that the logic of the former is essentially different from the logic of the latter, which is also the logic of the Kingdom of God. This is in accordance with the words of the prophet Isaiah, who put them in the mouth of God: "My thoughts," says the Lord, "are not like yours, and my ways are different from yours. As high as the heavens are above the earth, so high are my ways and thoughts above yours" (Isaiah 55:8–9). The logic of God is antisymmetric to the logic of the world, which means it is in a way still similar yet radically different. This difference is captured in the world of philosopher Anton Stres, who said that God is "differently different from things than things are different from each other." (Our translation) (Stres 1994, p. 414)

We can clearly see by analyzing the four layers of Western society that there is a movement from its interior to the exterior: the deductive logic was followed by inductive logic, inductive logic by the abductive logic and abductive logic by eductive logic. The inner personal life of Western society is, according to our analysis, slowly coming to the surface and changing its surface layer, which is part of the material world. This uppermost layer of Western society is seemingly becoming less and less worldly and more and more evangelical, as we have demonstrated on the individual cases from the structure of arguments, science and society. We interpreted this as a sign that Western society has taken the final step in conversion, enabling it to become the People of God, the heir of the Kingdom of Heaven proclaimed by Jesus of Nazareth.

However, we have further evidence that Western society has partly adopted double antisymmetry of personal life. In the last period of Western history, we have witnessed the thaw in relations between Western nations, as there is no longer any hatred between them, as was the case at the beginning of the 20th century, when the Old Continent was rocked by nationalisms and wars. Today, the European union enables peaceful coexistence between different nations and the free movement of people and goods, which brings prosperity to all. With the help of professions such as psychiatrists, we have learned to recognize behaviors resulting from trauma and other pathological forms. Minorities have become relatively well protected, we are working to eliminate hate speech, social security is better, and care for weaker and vulnerable members of society has increased. Today, in general, pluralism of opinion is the rule, which means that the world is no longer so black and white as it used to be. Western society is therefore well on its way to becoming the People of God, in which the whole takes precedence over parts—which means that individuals are organically connected in an organism in which everyone weeps with the grieving and rejoices with the happy—and in which the concrete dominates the abstract—which means that no one is overlooked or disregarded and that everyone is indispensable. Are these author's optimistic observations merely a consequence of the theory-ladenness of observations with the Christian worldview? Only time will show.

**Funding:** This research received no external funding.

**Institutional Review Board Statement:** Not applicable.

**Informed Consent Statement:** Not applicable.

**Data Availability Statement:** No new data were created or analyzed in this study. Data sharing is not applicable to this article.

**Conflicts of Interest:** The author declares no conflict of interest.

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
