# Peer review of "The Gospel’s Double Antisymmetry as the End-Point of the Development of Western Society"

_religions, doi:10.3390/rel13010080_

Round 1
Reviewer 1 Report
Of the many article submissions I have reviewed in my career, this particular piece is probably the strangest, and perhaps for this reason it quickened my attention. I am still not sure whether in the end the effort is altogether rewarded. The article is woven very meticulously, painstakingly, with apparent logic and patient deliberation, but it is not clear whether it makes up a cloth of any sort.
I shall proceed step by step, starting with the title:
A title should be an enticement to step in and a sketch of the argument. The title in question here is neither: it repels getting past it, and it is ostensibly not about the paper—which does not deal with the Gospels. Indeed this article is really not about religion at all, which shows up briefly and at the eleventh hour, in the last three or four paragraphs, to set a seal of indubitability on what is, by my estimate, a somewhat nebulous progress. Maybe the title means everything to the author; to the reader, it means nothing at all whether before or after perusing the article.
The author very authoritatively divides reality into four layers. Why four layers and not 9 or 86 or 723,5—we do not know. We just have to take it on trust that that is how reality works out, quite happily as it turns out, because it suits the author’s origami exercise of opposite and overlapping corners. The author says that he/she defends the thesis that they are four layers of reality, but the thesis is not defended; it is stated as a dogmatic premise. After that, it all gets a bit complicated—the four layers (which really are corners of reality) of material world, material life, lifeworld, and personal life looking at one another across multiple fold lines of anti-symmetry and double-anti-symmetry.
The rhetoric of the paper appears to use strict logic, but logic is given very rough handling throughout. It is pressganged to fit into the aforementioned fourfold scheme in which reality has kindly seen to arrange itself for our benefit. I am not sure the author grasps the idea of symmetry. Lines 72-76 claim that the Gospel’s reversal of first and last: this is not antisymmetry, since whether the first are the last or the last are the last, there is a first and last and the integrity of the system is preserved, and thus symmetrical.
This mention of religion and of Christianity in particular occurs in the introduction of the paper, and it is the last time the topic arises until the conclusion. Thus the main bulk of the paper is not about the Gospel, not about religion at all. Only its tacked-on conclusion is.
Logic suffers under the appearance of being the driving force. I cite this example, one among many. Lines 129-130: “the material world of corporeality is the starting point of our understanding of reality, for without observation there can be no other deeper perceptions of meaning.” The material world of corporeality is obviously not the starting point here, since the author claims it exists only by being perceived. Observation is therefore the starting point.
Likewise: the starting point is the part, not the whole. Line 133: “the precedence of parts (of components) over the whole (as a body): how can something be a part without being a part of something (a whole) is a mystery. In reality parts and whole are therefore perceived independently and diachronically.
“The deductive precedence of parts over the whole” (line 146): how can something that is deduced also precedent? If we deduce the parts from the whole (which is the deductive method) then how are the parts precedent? This is all very strange.
Paragraph 3.1.2: The author does not get the scientific method quite right. If it is deductive, it is not in the way of autocratic fiat. Theoretical physics does not impose on reality, the way an autocrat does. It formulates a hypothesis, which waits on the veto of observation and empirical verification.
Line 24: in science, “the concrete fact is proved by an abstract rule and vice versa”. This is nonsense. In science, there is a fact whether we say there is or not. The orbit of the planet Mars has the shape of an ellipse not because we say it does, but because, on careful observation and measurement, it turns out that it does.
I am just giving these examples as tokens of the often very wobbly logic used throughout the article. As I see, the article does not seek to demonstrate a reality, but to use the appearance of logic to cram reality into the four-fold scheme decreed by the legislating ontologist in control of the universe of this article.
The purpose and utility of the article is unclear throughout its dense development. The author seems to be playing with wooden blocks, shifting them here and there according to the game rules that are made up on the spot. The reader is very much not the wiser for it, and the topic of religion (or our understanding of the Gospels) totally left out of account. The reader has to wait until the conclusion to see how all this house of cards builds up to a ladder onto religion. It doesn’t. The author ends up positing that the true identity of a thing or person is its essence (Platonic theology), and that God knows us in our essence, and so the whole contains the parts, and the Kingdom of God triumphs over the pieces. How we arrive there by logical deduction; how the preceding origami interpolations lead to this irresistible conclusion: all this is extremely unclear, unstated, at the discretion of the author’s right to swoop in, change the rules of the game, rename a category, and establish the primacy of this or that.
Reviewer 2 Report
The article clearly states the argumentation. The methodology is adequately used to support the main hypothesis. The article´s structure is clear and logical. The content is coherent and engaging, as well.
Author Response
Thank you for your comments!
Reviewer 3 Report
This is a very interesting and provocative paper. I have two main suggestions: the first has to do with the way you frame the project, and the second is more substantive.
I. This paper is extremely ambitious, and (partly because of this) the philosophy you present here is of a very different kind than that of the Anglo/Germanic style that dominates English language philosophy. To open a space for your work to be heard and to establish the criteria on which you wish it to be judged, I would do two things:
(1) Very early in the paper (in the Abstract and Introduction) explain that this a “Prolegomena” or an introduction to a much larger project and that you will introduce many things that cannot be developed in full detail in an article length work. Then give a rationale for why a work of this complexity should be presented in an abbreviated from and not wait for a book.
(2) Make explicit that you are rejecting the way of doing philosophy that comes from the German Research University model. You are not identifying some small ‘hole’ in the published record that your work is filling, in the cumulative aggregation of knowledge. Rather you are attempting a synthetic and synoptic approach that attempts to gather together a new way of looking at reality as a whole, and in particular a way of understanding that emerges from an interdisciplinary approach.
II. The four main categories (Material world, Lifeworld, Material life, and Personal life) are difficult to follow and need work. You should consider changing the terminology or explaining the reasoning behind the repetition of the terms ‘life’ and ‘material.’ How are the material world and material life connected? How are lifeworld, material life, and personal life connected? Why does ‘life’ appear in three of the four categories?
These terms clearly have both epistemological and ontological valence, but what is the particular relation between these aspects? Do these concepts refer to different kinds of things? At times it seems that ‘material world’ refers to material things, ‘material life’ to living beings, ‘lifeworld’ to sentient beings, and personal life to ‘rational beings,’ but this is not clear. In particular ‘lifeworld’ does not seem to fit with the others very easily.
If it is true that there is an ontological hierarchy at work here (from merely material things up to persons), why would the way of understanding nonhuman things serve as a way of organizing human society? How is this possible?
- The idea that at the Last Judgement the entirety of God’s people will be involved in judging the individual person (880) seems implausible and theologically suspect for Christianity (see Matt 18, Luke 11:4, etc.). Defend this claim or remove.
- You end with a very optimistic claim that “our culture is becoming more and more a culture whose nature fits the personal life, proclaimed by the Gospel” (907). What does “our culture” refer to? Where do you see these positive signs at work?
Round 2
Reviewer 1 Report
The author certainly has put in an enormous impassioned work of justification and clarification. Though I still think that the article would benefit from more simplicity, and less scholasticism, it is no doubt very serious about its purpose the virtue of which cannot be faulted. I thus recommend the article be accepted in its revised form.